# Morphometric Changes of Osteocyte Lacunar in Diabetic Pig Mandibular Cancellous Bone

**DOI:** 10.3390/biom13010049

**Published:** 2022-12-27

**Authors:** Sheng Yao, Zhibin Du, Lan Xiao, Fuhua Yan, Saso Ivanovski, Yin Xiao

**Affiliations:** 1School of Mechanical, Medical and Process Engineering, Center of Biomedical Technology, Queensland University of Technology, Brisbane, QLD 4059, Australia; 2The First Hospital of Wuhan, Wuhan 430033, China; 3Australia-China Centre for Tissue Engineering and Regenerative Medicine, Queensland University of Technology, Brisbane, QLD 4059, Australia; 4Nanjing Stomatological Hospital, Medical School of Nanjing University, Nanjing 210008, China; 5School of Dentistry, The University of Queensland, Brisbane, QLD 4006, Australia; 6School of Medicine and Dentistry & Menzies Health Institute Queensland, Griffith University, Gold Coast, QLD 4222, Australia

**Keywords:** osteocyte, bone, diabetes mellitus, backscattered scanning electron microscope, morphology, Image J

## Abstract

Osteocytes play an important role in bone metabolism. The interactions of osteocytes with the surrounding microenvironment can alter cellular and lacunar morphological changes. However, objective quantification of osteocyte lacunae is challenging due to their deep location in the bone matrix. This project established a novel method for the analytical study of osteocytes/lacunae, which was then used to evaluate the osteocyte morphological changes in diabetic pig mandibular bone. Eight miniature pigs were sourced, and diabetes was randomly induced in four animals using streptozotocin (STZ) administration. The mandibular tissues were collected and processed. The jawbone density was evaluated with micro-CT. Osteocyte lacunae were effectively acquired and identified using backscattered electron scanning microscopy (BSE). A significantly decreased osteocyte lacunae size was found in the diabetic group. Using the acid etching method, it was demonstrated that the area of osteocyte and lacunae, and the pericellular areas were both significantly reduced in the diabetes group. In conclusion, a standard and relatively reliable method for analyzing osteocyte/lacunae morphological changes under compromised conditions has been successfully established. This method demonstrates that diabetes can significantly decrease osteocyte/lacunae size in a pig’s mandibular cancellous bone.

## 1. Introduction

It has been widely accepted that the osteocyte, as an orchestrator to regulate bone formation and resorption, plays a vital role in bone metabolism [1,2,3]. Osteocytes have a mechanosensory function and play a significant role in bone mineralization and mechanical conduction [4]. Recently, it has been reported that osteocyte interaction with the surrounding microenvironment can result in changes in the cellular and lacunar system morphology [5,6,7], which can be influenced by osteocyte maturation and disorders [1,8,9]. For example, morphological changes in osteocytes have been found in osteoarthritis, osteopenia, and osteoporotic bone [10,11].

Diabetes mellitus is a highly prevalent metabolic disease that causes nerve damage, arterial disease, and obesity. It also significantly influences bone metabolism and increases the risk of various bone-related disorders [12,13,14,15]. However, the changes in the jawbone in diabetes mellitus, especially changes in the osteocytes, which are considered a central regulator of bone homeostasis, remain largely unknown. Quantifying osteocyte lacunae is challenging because osteocytes are deeply embedded in the mineralized bone matrix. Backscattered electron (BSE) imaging has been used as an image tool to study bone microstructures [16] and osteocytes [17,18]. However, the artifact and mineral intensity in BSE usually prevents accurate osteocyte counting and lacunae measurement.

The current project aimed to establish a method for the study of osteocytes, and to use the established method to analyze the diabetic bone quality by measuring the osteocyte/lacunae size in the porcine mandibular cancellous bone. This project studied the changes in the jawbone in a large animal model of pig diabetes mellitus and, for the first time, investigated osteocyte morphological changes in the diabetic porcine mandibular bone using the technique developed in this study using a scanning electron microscope with backscattered electron imaging (SEM-BSE) and acid etching methods.

## 2. Materials and Methods

### 2.1. Pig Diabetes Model and Mandibular Bone Sample Preparation

All the animal procedures were approved by the Nanjing University Animal Ethics Committee, and the experiments were performed in biosafety level 3 (BSL-3) facilities at the Department of Comparative Medicine at the General Hospital (Nanjing 210002, China). Eight healthy Guangxi Bama miniature pigs, one-year-old with a body weight of around 50 kg, were randomly sourced and divided into diabetes and control sham groups. After a fortnight of adaptive feeding, pigs in the diabetes group received an ear vein injection of Streptozotocin (STZ, Sigma, 150 mg/kg, diluted in 9.5 mL/mg sterile saline) to induce diabetes, following a previously established protocol [19]. The diabetes model was validated via the continually elevated fasting blood glucose levels [20], which are shown in Figure 1. 48 days after STZ induction, the blood glucose of the pigs in the diabetes group was significantly higher than the control group, and the high level of blood glucose was maintained for 90 days. After verification of the diabetes model, the animals were sacrificed, and the mandibular bone was collected and fixed in 10% formalin. The molar teeth and associated bone were segmented and embedded in resin (Technovit 7200 Heraeus Kulzer GmbH, Wehrheim, Germany) according to the manufacturer’s instruction. Resin embedded sections of 1 cm thickness were obtained by sequentially polishing (starting from 1200 of mesh size of sandpaper and then using diamond particle suspensions of 6, 1, 0.25, and 0.05 µm sequentially). The sample sections were coated with gold before observation using BSE. Four samples were prepared from one animal, and one was selected randomly for observation. A total of four sections in each group were used for backscatter scanning electron microscopy (BSEM) observation.

### 2.2. BSE Image Acquisition for the Analysis of Osteocyte/Lacunae

BSE images of osteocyte/lacunae were obtained using Tescan MIRA 3 Electron Microscope system (Tescan, Brno, Czech Republic). In order to exclude the influence factor of the root tip on the osteocyte morphology, the cancellous bone at least 5 mm away from the molar teeth root tip was selected for image acquisition. The magnification of BSE images was set at 200 times to capture enough bone area to avoid selection bias. The electron beam energy was set at 15 kV with 13 beam intensity and 15 mm working distance. The pixel resolution corresponding to a magnification of 200 was 1.35 (micrometer/pixel). Three images of different parts in each bone sample were captured randomly in the BSE visual capture field.

### 2.3. Post-Acquisition Image Processing

A standard BSE image process workflow was established for post-acquisition image processing to develop a standardized osteocyte identification and quantification method. All the BSE images were processed and analyzed using this standard workflow via the freeware Image J (Schneider et al., 2012). This workflow has three steps: binary segmentation, noise removal, and porosity identification. Figure 2 summarizes the entire adapted processing workflow.

The purpose of the “binary segmentation” was to convert an original BSE image into a binary image. Global thresholding is an image segmentation method that replaces a pixel in a greyscale image with a black pixel if the image intensity is lower than some fixed constant (the threshold), or a white pixel if the image intensity is higher than that constant. We used “intermodes global” thresholding (Figure 2B) to identify the intensity values of the original BSE image, and the histogram of the image intensity value was given by image J software. This histogram included two peaks (P1 and P2 in Figure 2B). The value of the peak top could be captured and the value of the middle point could be calculated when the mouse cursor was slowly slid from left to right to obtain the value of the middle point. With the increase in the image amount, the selection bias could be controlled. Our original BSE images were 16-bit images. The pixel values were unchanged with the change in brightness and contrast. This method of image thresholding has been suggested in the Image J user guide (https://imagej.nih.gov/ij/docs/guide/user-guide.pdf, accessed on 24 March 2019). As we used “intermodes global” thresholding to identify the intensity values of the original BSE image, the pixel greyscale histogram of all the included original BSE images appeared to have the typical two peaks. If the greyscale histogram of the image did not show the typical two peaks, it was excluded in our research. This ensured the quality of the original BSE Image had a sensible brightness/contrast quality, thus ensuring the thresholding.

After the threshold operation, the grey scale of the original BSE image was converted into a binary image, in which black represented objects and white was the background. This step offered standard images for us to identify the osteocytes for analysis.

The “noise removal” step allowed for the precise identification of osteocyte structures and thus improved the accuracy of the binary images. A series of operations aimed at removing artifacts (holes and noise) was carried out using Image J tools. This procedure enhanced the quality of the binary images so that the edge of the objects in the binary images was more precise and sharper.

The “porosity identification” was used to differentiate the osteocytes from the porous-like objects. Figure 3 summarizes the entire adapted processing. To set an identification range of osteocytes, we first tried to calculate the area of all objects in every BSE picture. After the area calculation of all objects, we obtained the area histogram, including all the objects. These objects might include noise objects, osteocytes, and other canals such as blood vessels, Haversian canals, or bone marrow cavities. In fact, there is no lower limit on the possible size of a lacuna cross-section because the 2D sections of lacunae are investigated. However, from the observation of the histogram picture, we found that the number of objects less than 8 µm^2^ increased sharply with the decrease in area. This indicates that an object with an area of less than 8 µm^2^ may significantly increase the additional noise. Therefore, these uncertain objects (less than 8 µm^2^ in the histogram) were defined as noise and excluded in the osteocyte analysis. Moreover, the areas from 100 µm^2^ to 500 µm^2^ were segmented to create separate serial images every 50 µm^2^. These were compared with the original image to determine the most suitable threshold area value to define the osteocyte lacunae. Objects larger than 100 µm^2^ decreased sharply and were easy to observe. With the area increasing from 100 µm^2^ to 500 µm^2^, if we found an object was obviously falsely recognized as an osteocyte, we would stop the comparing process and the previous value was recorded as the maximum size criterion. This process was repeated 24 times for all images (three images for each sample, four samples for each group) to obtain an average value of 280 µm^2^, which was set as the maximal threshold for defining osteocyte lacunae. Areas larger than 280 µm^2^ were identified as other canals, such as blood vessels, Haversian canals, or bone marrow cavities.

However, some small cracks were wrongly identified as osteocyte lacunae, which needed to be removed even after the above classification.

The parameter of Circ. in Image J software was used as a filter to remove the incorrectly identified small cracks. Circ. is a parameter given by Image J to describe the circle level of one object. The definition of Circ. parameters follows the user guide of Image J on the official website. (https://imagej.nih.gov/ij/docs/guide/user-guide.pdf, accessed on 24 March 2019). $Circ.\;:\;4\pi \times \frac{\left[Area\right]}{\;{\left[Perimeter\right]}^{2}}$ with a value of 1.0 indicating a perfect circle. As the value approaches 0.0, it indicates an increasingly elongated shape. The parameter of Circ. of all objects is demonstrated in the histogram (Figure 4D). From 0 to 0.5, the objects are segmented into a separate image every 0.1 to compare with the original image so as to precisely define the actual value of the osteocyte lacunae and obtain the threshold value. This process was also repeated 24 times. Finally, the average threshold value of 0.34 was set as the minimal boundary line of the osteocyte lacunae. Objects with a parameter of Circ. less than 0.34 were defined as small cracks and were removed. Figure 4B shows that these cracks were marked by red circles. Figure 4C shows the results of segmentation. After the porosity identification, osteocyte lacunae were precisely identified, and morphological data acquisition was undertaken.

### 2.4. Acquisition of Osteocyte Lacunae Morphometric Data

Parameters such as the lacunae area, lacunae perimeter, lacunae roundness, and aspect ratio of the lacunae were automatically calculated by Image J software. The definition of these parameters follows the user guide of Image J on the Image J website (https://imagej.nih.gov/ij/docs/guide/user-guide.pdf, accessed on 24 March 2019). The density of lacunae was also analyzed according to the formula provided below.

Area: Area of selection in square unitsPerimeter: The length of the outside boundary of the selectionAspect ratio: The aspect ratio of the particle’s fitted ellipse, [Major Axis]/[Minor Axis]$Roundness:4\times \frac{\left[Area\right]}{\pi \;\times \;{\left[Major\;axis\right]}^{2}}$ or the inverse of the aspect ratio

From 0 to 1. As the value approaches 1.0 to 0.0, it indicates an increasingly elongated shape.
$$Density\;of\;lacunae=\frac{Number\;of\;lacunae}{Total\;area-area\;of\;other\;canals}$$

### 2.5. Method Reproducibility Analysis

To examine the reproducibility and robustness of the established method, a total of 9 images in the control group and 12 images in the diabetes group were analyzed following the same method. The images originated from the same sample but were taken in different experimental sessions.

### 2.6. BSE Image for Acid-Etched Resin Embedded Samples

To further confirm the results from the BSE images process workflow, the osteocytes in the acid-etched resin embedded samples were manually collected for measurement. The same samples were acid-etched with 9% phosphoric acid for 15 s, then immersed in bleach for 5 min, followed by washing with distilled water and coated with gold according to the previously published method [21]. BSE was also used to capture visible resin-casted osteocytes and lacunae in the same condition described above. The focusing and image acquisition process for each image were completed within 30 s. A total of 100 osteocytes in each group were randomly collected for the analysis. The area of the osteocyte and surrounding lacunae were measured to compare between two groups.

### 2.7. Micro-CT Data Acquisition for Trabecular Bone

To recognize the trabecular bone change in the diabetic group, the data of the trabecular bone micro-CT were also analyzed after the data acquisition. Micro-CT data of the jaw trabecular bone were obtained using µCT50 (µCT50; Scanco Medical AG, Bassersdorf, Switzerland) after jawbone sample preparation. The bones were scanned at an energy of 70 kV and intensity of 114 µA, with 300 ms integration time, resulting in a 16 µm isotropic voxel size. The pixel resolution is 0.016 (mm/pixel). A total of six samples of the jawbone, including three in the control group and three in the diabetic group, were prepared for micro-CT scanning. A total of three cubes (0.344 cm × 0.344 cm × 0.344 cm) of bone within the tooth root bifurcation area were selected for analysis in every sample. The analysis parameter setting for the micro-CT data is shown in Figure 5.

### 2.8. Statistical Analysis

All of the statistical analyses were performed using GraphPad Prism 7.0. (GraphPad Software, San Diego, CA, USA). Data following normal distribution are expressed as mean ± SD, and error bars represent SD. Interquartile range (IQR) show data following a skewed distribution. The Mann–Whitney test was used to compare the rank. The Kolmogorov–Smirnov test was used to compare the difference in distribution. *p* < 0.05 was considered a statistical difference.

## 3. Results

### 3.1. Establishment of a Diabetic Pig Model

In our study, after half a month of adaptive feeding, the pigs in the diabetes group received an ear vein injection of streptozotocin (STZ, Sigma, 150 mg/kg, diluted in 9.5 mL/mg sterile saline) to induce diabetes [19]. The diabetes model was validated via the continually increased fasting blood glucose levels compared with the control group, as shown in Figure 1.

### 3.2. Establishment of BSE Image Processing Workflow

Figure 2 shows the workflow for the BSE image process for correctly identifying osteocytes. BSE images were binarized by internode global thresholding of the Image J software. Clear images were obtained by filtering out artifacts to identify the lacunar osteocytes. Porosity identification was carried out to separate osteocyte lacunae from other objects. The midpoint line of the histogram of the image intensity value was used as the thresholding line for the image threshold.

Following the BSE image processing workflow on Image J, the osteocyte lacunae in the BSE images were successfully distinguished from the image noises and other cavities, such as blood vessels and Haver’s canals.

### 3.3. Morphometric Changes of Osteocyte Lacunar in Diabetic Pig Mandibular Cancellous Bone

Based on the method reported above, the diabetes group significantly decreased the lacunae area and its perimeter in the mandibular bone compared with the control group (*p* < 0.0001) (Figure 6A,D). The differences in area and perimeter distribution are also shown in Figure 6B,E. Figure 6C shows that the area of osteocyte lacunae in the diabetes group decreased compared with the control group. Figure 6F shows that the perimeter of the osteocyte lacunae in the diabetes group decreased compared with the control group. The result indicates that the size of osteocyte lacunae in the diabetes group was significantly decreased in the diabetes group. However, the roundness, the aspect ratio, and the distributions of the osteocyte lacunae did not show any significant difference between the two groups (*p* > 0.05) (Figure 7). In addition, the density of the osteocyte lacunae between the two groups did not present any difference either (*p* = 0.3777) (Figure 8).

### 3.4. The Reproducibility and Robustness of Our BSE Image Analysis Method

To examine the reproducibility of the established analysis method, images of the same samples were taken during different experimental sessions, which were analyzed using our method, and the analysis results of the images taken at two sessions (analysis 1 and analysis 2) were compared. As shown in Figure 9, the results of analysis 2 were in accordance with analysis 1, which demonstrates the reproducibility and robustness of the established BSE image analysis method.

To verify the obtained results following the workflow process, we also did measurements by manually selecting osteocyte lacunae of the acid-etched samples. After acid-etching with 9% phosphoric acid, the osteocyte lacunae were imaged using BSE. The pericellular area was clearly shown in the high magnification of the BSE image (Figure 10A,D). Figure 10B,E shows the binary picture after processing, and the pericellular area can be observed. The size of osteocytes was significantly reduced in the diabetes group compared with the control group (*p* < 0.001) (Figure 10G), and the related lacunae and the pericellular area in the diabetes group were also significantly diminished (*p* < 0.001, *p* < 0.05, respectively) (Figure 10H,I).

### 3.5. Morphometric Changes of Trabecular Bone in Diabetic Pig Mandibular Cancellous Bone

The micro-CT results suggest that the trabecular thickness (Tb. Th), trabecular separation/spacing (Tb. Sp), and trabecular surface area under the same volume (BS/BV). The trabecular number (Tb. N) and some other indicators, including structure model index (SMI) and the bone volume under the same total volume (BV/TV) show no significant difference between the diabetic group and the control group (as shown in Figure 11).

## 4. Discussion

Osteocytes are recognized as key mechano-sensors and transducers in bone. The osteocyte morphology changes may affect its role in mechanical induction and bone adaptation [4,10]. It has been demonstrated in vitro that spherical osteocytes are much more mechanosensitive than flat ones [8]. Morphological alteration of osteocytes has been found in different types of bone diseases. Van Hove [10] showed that osteoarthritic bone presents relatively elongated osteocytes, and osteopenia bone has relatively round osteocytes, while osteopetrosis bone has relatively small and round osteocytes. The previous study also claimed that infantile bone with large osteocyte lacunae indicates a developmental phenomenon rather than osteocytic osteolysis [22]. Hypermineralization and high osteocyte lacunar density have been found in osteogenesis imperfecta type V bone [23]. Previously, we also demonstrated the morphometric change of osteocytes in osteoarthritis [11]. These morphological changes of osteocytes may reflect bone metabolic changes in these bone diseases. Understanding these cell morphological changes may pave the way for new progress in preventing and treating bone disease [24].

However, osteocytes remain among the most challenging to study due to their inaccessible location within the mineralized bone matrix. Numerical illustration and comparison of osteocyte lacunae are complicated. Backscattered electron (BSE) imaging has been used as a reliable image tool to study bone microstructures [16] and osteocytes [17,18]. However, the artifact and mineral intensity in BSE usually prevent the accuracy in osteocyte counting and osteocyte lacunae measurement. Therefore, establishing a simple, repeatable, and robust method to analyze osteocyte properties using BSE remains challenging. The purpose of this project was to develop a technique using Image J, a Java-based software with an open resource [25], to identify artifacts and reduce background noise to quantify the morphological changes of osteocytes/lacunae using BSE images acquired by the scanning electron microscopy and backscattered electrons imaging (SEM-BSE).

We established a robust, normative, and simple strategy for BSE image processing and analysis workflow so that the osteocyte lacunae could be identified from background BSE image objects, facilitating the measurement of the osteocyte morphometric properties. This method enabled the study of the osteocyte morphology by removing minor artifacts (cracks and noises) from the osteocyte lacunae and increasing the accuracy of osteocyte counting and lacunae measurement. This workflow includes three steps. The first step, named “BINARY SEGMENTATION”, sets the standard for all BSE images to be converted to binary images. Step 2, called “NOISE REMOVAL”, is set to remove little noises to increase the accuracy of the BSE images. Step 3, named “POROSITY IDENTIFICATION”, generated a normative strategy to identify the osteocyte lacunae. The workflow provided a novel standard and simple method for osteocyte lacunae measurement. It utilized two simple parameters in Image J software to identify osteocyte lacunae after image binary segmentation and noise removal. It enabled a robust method for numerous measurements without the selection bias originating from manual measures. We think this is a robust and convenient method option for osteocyte identification and measurement.

The range setting of osteocytes is changeable with the change in different samples from different animals. However, this setting procedure is reproducible and can be used for this kind of image processing, which is especially useful in the geometric properties comparing numerous osteocytes between groups. Our reproductivity analysis (analysis 2) used nine images in the control group and 12 images in the diabetes group. All of these images originated from the same sample surface, but were taken during another experiment session. We analyzed these BSE images following the same method. The results coincide with the previous analysis (analysis 1) and thereby support the reproducibility and robustness of our approach.

To verify the BSE image processing workflow results, we manually measured the osteocytes in acid-etched resin-embedded samples. In our previously published papers [11,17], we demonstrated that backscattered SEM and acid etching methods will enable the study of the osteocyte morphological changes and lacuna–canalicular network in resin-embedded bone samples. Using the acid etching method, we also showed the changes in mineral density around the osteocyte body, which form the foundation for the techniques developed in this paper to measure the lacunae area. The acid will etch the minerals around the canaliculi to enable the view of the canaliculi network. Similarly, the acid will also etch the minerals around the osteocyte body. Due to the progressing decrease in mineral contents close to the osteocyte body, the lacunae area can be pictured as shown in Figure 10 in this study with the appropriate acid concentration and etching time. Compared with the cell body, dendrites, and high-density bone matrix, the demineralized areas around the osteocyte body, the lacunae area, will show a dark zoom under the backscattered SEM. In this study, we reported, for the first time, measuring the lacunae area using the acid etching method and backscattered SEM. We believe that our study will provide some information on how to analyze the osteocyte/lacunae morphological changes. Using a diabetes disease model, we validated the measurement of osteocytes and lacunae.

The manual and automated methods showed consistent results, which confirmed that our workflow strategy using BSE images and Image J software analysis is reliable. However, the current plan is still limited to only 2D analysis, as our results were based on 2D images of BSE. Information obtained from 2D images may lose some of the features of 3D objects.

The osteocyte morphological alterations in diabetic conditions were investigated using the established methodology. It was demonstrated that the size of osteocytes decreased significantly in the diabetes group. Osteocytes are recognized as not being homogeneous cells. Osteocytes can be divided into three groups according to their metabolic activities [26], whereby large cells indicate active metabolism, small cells indicate inactivity, and empty cavities indicate osteocyte necrosis. It is also suggested that lacunae become smaller with increasing age [27]. Our experiment showed that diabetes might decrease metabolic activity and alter mineralization. Indeed, previous research has found that in a diabetes animal model, the apoptosis of osteocytes is increased [28]. It has been suggested that failure to maintain normal blood sugar levels in diabetic patients may impair bone response to mechanical load [29]. In diabetic patients, the bone microstructure seems disturbed and may lead to bone fragility [30,31]. Some studies have revealed that osteocytes can modify their microenvironment by depositing and resorbing bone around the lacunae [32,33,34]. There is also a study claims that deterioration in the canalicular network with age reduces the connectivity between osteocytes and between osteons/interstitial tissue, which affects the supply of nutrients to osteocytes, degrades their mechanosensitivity, and contributes to increased bone fragility [35]. Our results suggest some degree of osteocyte metabolism changes. However, the exact relationship between osteocyte morphology and bone architecture is complicated and warrants further investigation. The shape of the osteocytes corresponds to the pathological state of bone tissue in the diabetic group, but knowing what the cause and the effect are is still worth further investigation.

Due to the small number of our sample, our micro-CT data show no significant difference between the diabetic group and the control group. Further research on a large sample size is suggested to verify whether there are changes on trabecular bone quality and the relationship with the osteocyte morphologic changes.

In conclusion, this study successfully established a standard and reliable method for analyzing osteocyte/lacunae morphological changes under compromised conditions. This method demonstrates the effects of diabetes mellitus on osteocytes/lacunae morphology, showing that diabetes can significantly decrease and osteocyte/lacunae size in a pig’s mandibular cancellous bone.

## Figures and Tables

**Figure 1 biomolecules-13-00049-f001:**
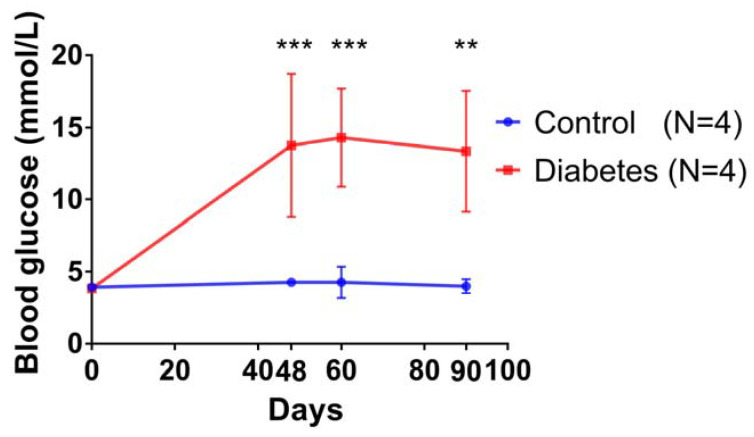
Fasting blood glucose of the pigs in two groups. Here, 48 days after STZ administration, the blood glucose of pigs in the diabetes group rose higher than average and was kept at a high level until 90 days (** *p* < 0.01; *** *p* < 0.001).

**Figure 2 biomolecules-13-00049-f002:**
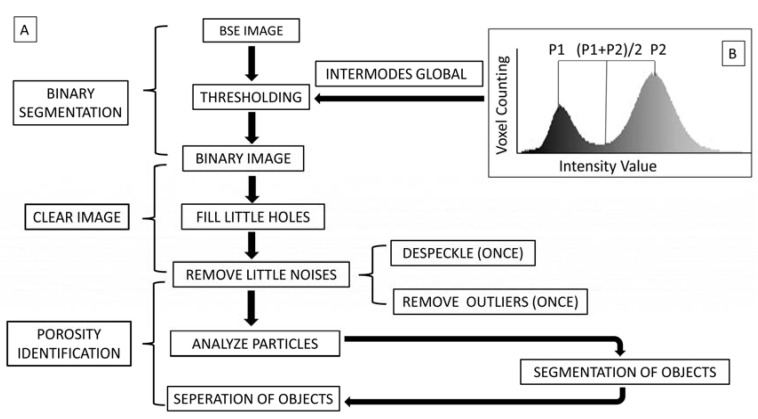
(**A**) BSE image processing and analysis workflow. Step 1. BSE Images were binarized by the internode global thresholding of Image J software. Step 2. Clear images were obtained by filling little holes and removing little noises to identify lacunar osteocytes. Step 3. Porosity identification was fulfilled to separate osteocyte lacunae from other objects. (**B**) The midpoint line of the histogram of image intensity value was used as the thresholding line for the image threshold.

**Figure 3 biomolecules-13-00049-f003:**
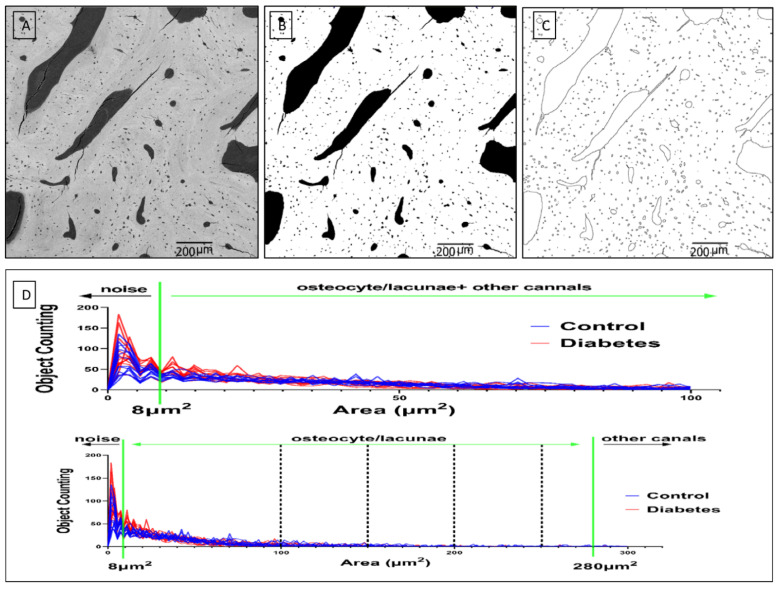
(**A**) The original image. (**B**) The binary image. (**C**) A picture of all objects has been demonstrated for object segmentation. (**D**) Classification criteria for object segmentation. The area histogram of all objects is shown by the location of all the objects. Objects less than 8 µm^2^ in size are defined as noises. To obtain the actual maximal area boundary line of the osteocyte lacunae, objects with an area between 100 µm^2^ and 500 µm^2^ were segmented into a separate image every 50 µm^2^ to be compared with the original image to make sure of the exact value to define the osteocyte lacunae area. This process is repeated 24 times because three images are selected from every sample. There are four samples in each group used for observation by BSE. Finally, 280 µm^2^, the average value of 24 images, is set as the maximal boundary line of the osteocyte lacunae. Objects with an area larger than 280 µm^2^ were defined as other canals, which include blood vessels, Haver’s channels, and others.

**Figure 4 biomolecules-13-00049-f004:**
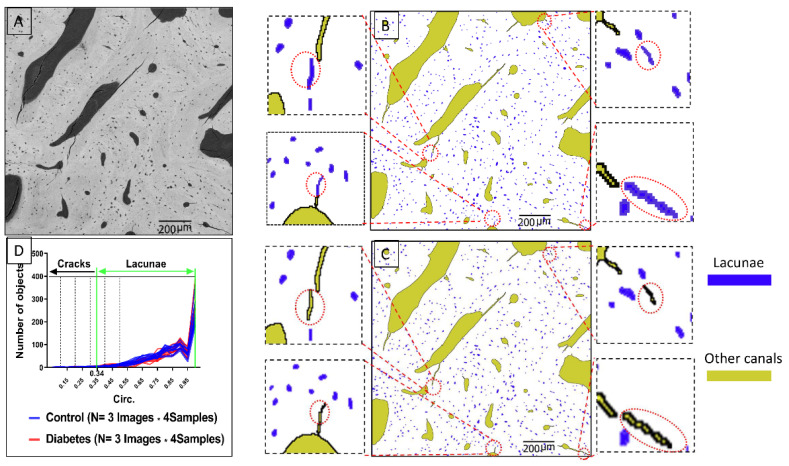
(**A**) The original image. (**B**) The image with small cracks that have been wrongly identified as osteocyte lacunae after the segmentation of objects in BSE images. These cracks are marked in the red circle. (**C**) The small cracks have been correctly removed from the osteocyte lacunae to be other canals. These cracks are marked in the red circle. (**D**) Classification criteria for the removal of small cracks from the osteocyte lacunae. A histogram of the parameter of Circ. of all objects is demonstrated. To obtain the specific minor Circ. boundary line of the osteocyte lacunae, objects of Circ. from 0 to 0.5 are segmented to a separate the images every 0.1 to compare with the original image to make sure which value is the exact value for defining the osteocyte lacunae. This process is repeated 24 times because 24 images are selected from two groups, as shown in picture D. Finally, 0.34, the average value, is set as the minimal boundary line of osteocyte lacunae. Objects with the parameter of Circ. less than 0.34 are defined as small cracks.

**Figure 5 biomolecules-13-00049-f005:**
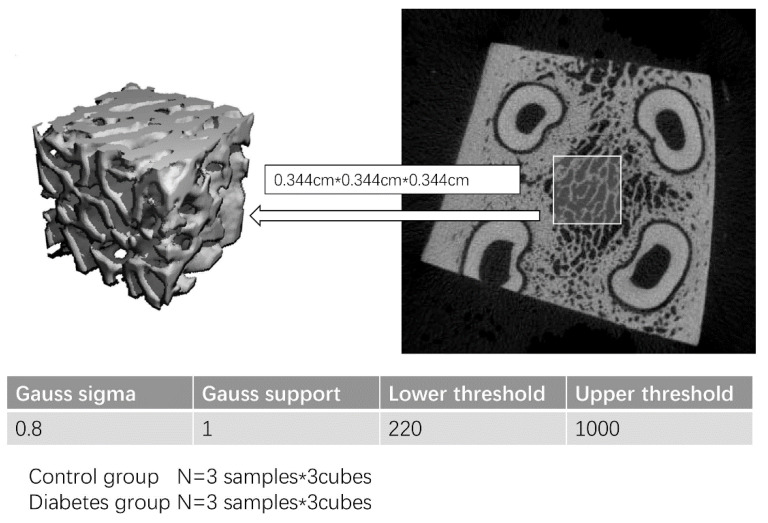
Micro-CT analysis of two groups. A total of six samples of the jaw bone, including three in the control group and three in the diabetes were prepared for the micro-CT scan. Three cubes (0.344 cm × 0.344 cm × 0.344 cm) of bone within the tooth root bifurcation area were selected for analysis in every sample.

**Figure 6 biomolecules-13-00049-f006:**
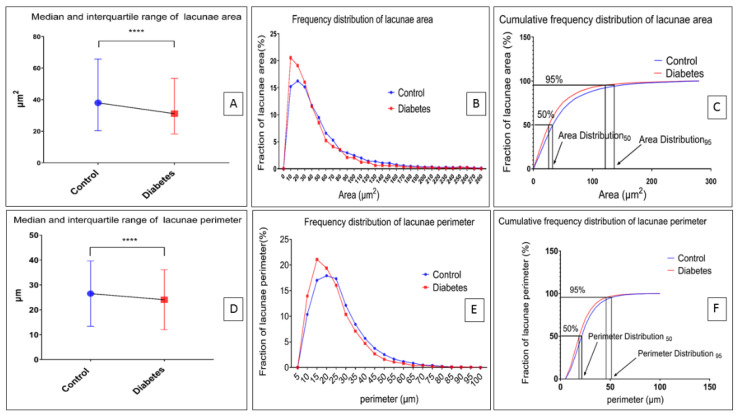
Osteocyte lacunar morphometric properties of the area and perimeter. (**A**) Properties of area. The interquartile range (IQR), range, and median lacunae area (µm^2^). Control group vs. diabetes group: IQR = 20.293–65.741 vs. 18.262–53.499; range = 9.049–279.401 vs. 9.131–278.515; median = 38.006 vs. 31.15 (*p* < 0.0001, Mann–Whitney test). (*p* < 0.0001, Kolmogorov–Smirnov test). (**B**) Distribution line of the area in diabetes and control groups. (**C**) Lacunae area cumulative frequency distribution. The cumulative frequency distribution of the control is below that of the diabetes group. (**D**) Properties of the perimeter. The interquartile range (IQR), range, and median lacunae perimeter (µm). Control group vs. Diabetes group: IQR = 16.87–32.63 vs. 15.32–29.61; range = 9.513–97.55 vs. 9.555–96.62; median = 23.62 vs. 21.27 (*p* < 0.0001, Mann–Whitney test). (*p* < 0.0001, Kolmogorov–Smirnov test). (**E**) Distribution line of the perimeter of lacunae in the diabetes group and control group. (**F**) Perimeter cumulative frequency distribution of lacunae. The cumulative frequency distribution of the control is below that of the diabetes group. (**** *p* < 0.0001).

**Figure 7 biomolecules-13-00049-f007:**
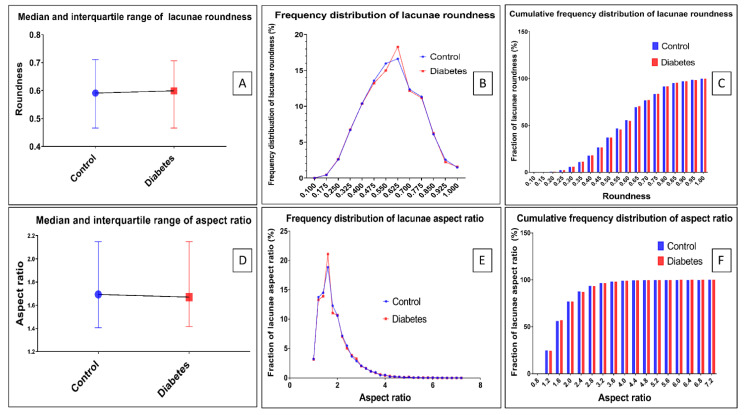
Osteocyte lacunar morphometric properties of the roundness and aspect ratio. (**A**) Properties of roundness. The interquartile range (IQR), range, and median lacunae area (µm^2^). Control group vs. diabetes group: IQR = 0.466–0.7113 vs. 0.466–0.706; range = 0.142–1 vs. 0.164–1; median = 0.5905 vs. 0.5899 (*p* = 0.9439, Mann–Whitney test). (*p* = 0.2368, Kolmogorov–Smirnov test). (**B**) Distribution line of lacunae roundness in the diabetes group and control group. (**C**) Cumulative frequency distribution of lacunae roundness. No noticeable difference can be seen between the two groups. (**D**) Properties of aspect ratio. The interquartile range (IQR), range, and median lacunae aspect ratio. Control group vs. diabetes group: IQR = 1.406–2.148 vs. 1.416–2.148; range = 1–7.051 vs. 1–6.089; median = 1.867 vs. 1.869 (*p* = 0.9431, Mann–Whitney test). (*p* = 0.2368, Kolmogorov–Smirnov test). (**E**) Distribution line of the aspect ratio in the diabetes group and control group. (**F**) Cumulative frequency distribution of the lacunae aspect ratio. No obvious difference can be seen between the two groups.

**Figure 8 biomolecules-13-00049-f008:**
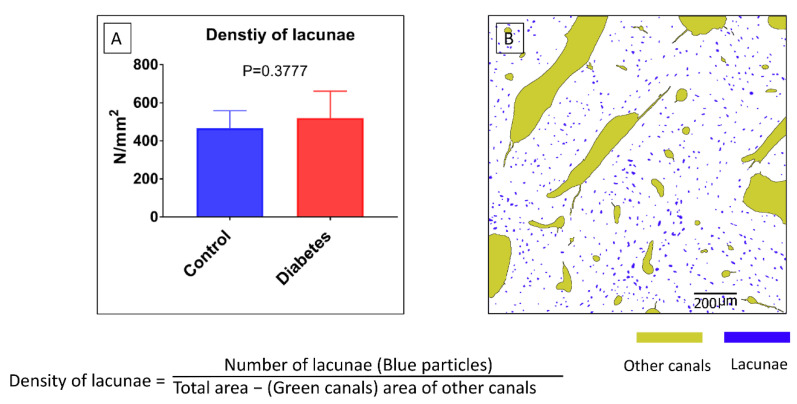
Osteocyte lacunar morphometric properties of the density. (**A**) The formula for density calculation is given. There is no significant difference between the two groups. (**B**) The segmentation of lacunae and other canals is shown.

**Figure 9 biomolecules-13-00049-f009:**
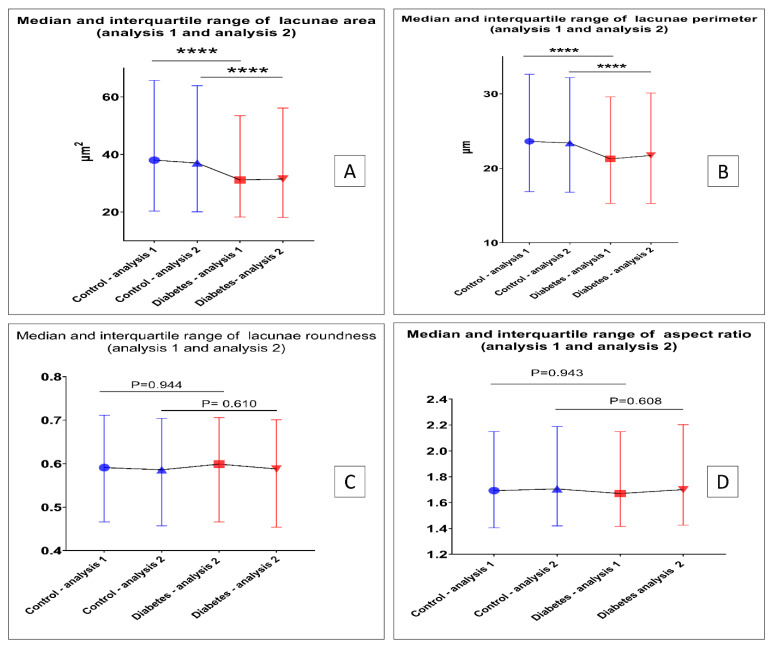
Properties of the area, perimeter, roundness, and aspect ratio from analysis 1 and analysis 2. (**A**) The area of osteocyte/lacunae in both analysis 1 and analysis 2 shows a significant difference between the control group and the diabetes group (*p* < 0.0001, Mann–Whitney test). (**B**) The perimeter of osteocyte/lacunae in both analysis 1 and analysis 2 shows a significant difference between the control group and the diabetes group (*p* < 0.0001, Mann–Whitney test). (**C**) The roundness of the osteocyte/lacunae in both analysis 1 and analysis 2 shows no significant difference between the control group and diabetes group (analysis 1, *p* = 0.944, Mann–Whitney test; analysis 2, *p* = 0.610, Mann–Whitney test). (**D**) The perimeter of aspect ratio in both analysis 1 and analysis 2 shows no significant difference between the control group and the diabetes group (analysis 1, *p* = 0.943, Mann–Whitney test; analysis 2, *p* = 0.608, Mann–Whitney test) (**** *p* < 0.0001).

**Figure 10 biomolecules-13-00049-f010:**
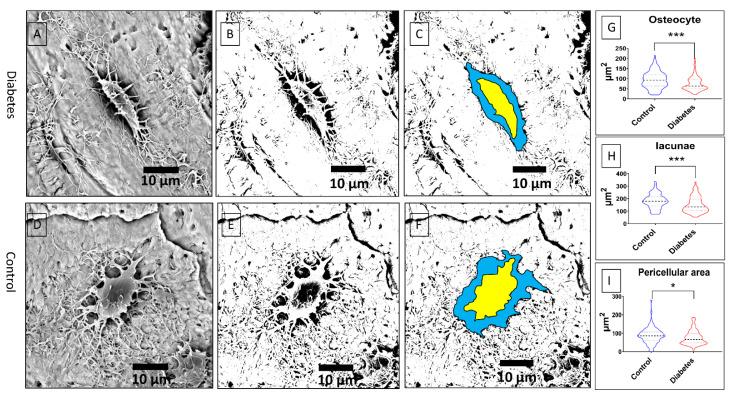
(**A**,**D**) The osteocyte lacunae morphology in acid-etched samples using BSE in two groups. The pericellular area was demonstrated in the high magnification of the BSE image. (**B**,**E**) The binary picture after thresholding; the edge of the peri-lacunae area can be observed obviously. (**C**,**F**) The location of osteocytes and lacunae for analysis. (**G**) The size of osteocytes is significantly decreased in the diabetes group (*p* = 0.0003 < 0.001, Mann–Whitney test). (**H**) The lacunae area is considerably reduced in the diabetes group (*p* = 0.0002 < 0.001, Mann–Whitney test). (**I**) The pericellular area is reduced to a small extent in the diabetes group. This difference also has a statistical significance (*p* = 0.0164 < 0.005, Mann–Whitney test; * *p* < 0.05; *** *p* < 0.001).

**Figure 11 biomolecules-13-00049-f011:**
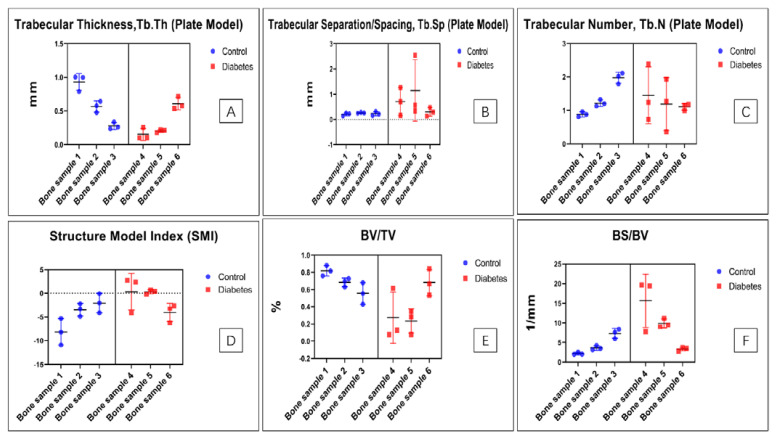
The micro−CT results show no significant difference between the diabetic group and the control group. (**A**) The trabecular thickness shows no significant difference (nested *t*-test, *p* = 0.3200). (**B**) The space between the trabecular bone has no significant difference (nested *t*-test, *p* = 0.0751). (**C**) The trabecular number shows no significant difference (nested *t*-test, *p* = 0.7655). (**D**) The structure model index shows no significant difference (nested *t*-test, *p* = 0.2155). (**E**) The bone volume under the same total volume shows no significant change (nested *t*-test, *p* = 0.1485). (**F**) The trabecular surface area under the same volume shows no significant change (nested *t*-test, *p* = 0.2465).

## Data Availability

Not applicable.

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
