# Peer review of "Morphometric Changes of Osteocyte Lacunar in Diabetic Pig Mandibular Cancellous Bone"

_biomolecules, 2022, doi:10.3390/biom13010049_

Round 1

Author Response

Dear Editors,

Thank you for your letter and for providing us with the reviewers’ comments on our manuscript. Those valuable comments are constructive for revising our paper, providing crucial guiding significance to our research. We have carefully revised our manuscript accordingly and believe that the revised manuscript can meet the criteria for acceptance. The changes are marked in blue and detailed in the point-to-point responses below.

Comments from reviewer 1:

  1. The paper introduces an image processing and analysis workflow to analyze osteocyte lacunae number and shape from BSE images. The authors are aware of the difficulties of such an approach, as they state in the discussion section in the paragraph starting from line 321: “But the artifact and mineral intensity in BSE usually prevent the accuracy in osteocyte counting and osteocyte lacunae measurement. Therefore, establishing a simple, repeatable, and robust method to analyze osteocyte properties using BSE remains challenging.” But their following claim that “We established a robust, normative, and simple strategy for BSE image processing and analysis workflow so that the osteocyte lacunae could be identified from background BSE image objects, facilitating the measurement of osteocyte morphometric properties.” is not backed by the results of the paper. Nowhere in the paper the robustness of the approach was checked. Most importantly, the authors did not investigate, how reproducible their approach is, if images from the same sample surface but taken during different experimental sessions are analyzed.

RE: We thank the reviewer for this valuable comment. The claim for robustness is not very appropriate for this study and we have revised the claim accordingly.   

The principle of our method is image processing based on image J software and how to set a reasonable osteocytes range for numerous comparisons between different groups. The range value is changeable with the change of different samples from various animals or different experimental sessions. However, this strategy is reproducible and can be used for this kind of image processing and be useful in the geometric properties comparison of numerous osteocytes.

To examine the reproducibility and robustness of our method, a total of 9 images in the control group and 12 images in the diabetes group were further analyzed based on the same strategy. All these images originated from the same sample surface but were taken during different experiment sessions. The results from the analysis (analysis 2) are in accordance with the results of the previous analysis (analysis 1), which have been presented in the revised manuscript (revised Figure 9). This can strongly support the reproducibility and robustness of our approach. We described these analyses and comparison details in the revised manuscript. (line242- line246; line404- line410)  

  1. It is clear that all geometric properties (the size and shape) of the lacunae depend on the binarization. This binarization itself depends on the location of the position of the two intensity peaks used in the internodes-algorightm. But how well are these controlled? They naturally change with changed brightness/contrast settings during different measurement sessions, unless brightness/contrast (and, thus, also sample current) is precisely controlled. Nothing is said in the manuscript how such changes influence the results.

RE: We thank the reviewer for this valuable comment. Global thresholding is an image segmentation method that replaces a pixel in a greyscale image with a black pixel if the image intensity is less than some fixed constant (the threshold) or a white pixel if the image intensity is greater than that constant. We used “intermodes global” thresholding to identify the intensity values of the original BSE image, and the histogram of the image intensity value was given by image J software. This histogram includes two peaks, and the value of the peak top can be captured as we slide the mouse cursor slowly from left to right. It is therefore not difficult to get the value of the middle point. And with the increasing image amount, the selection bias can be controlled. Our original BSE images are 16-bit images. The pixel values are unchanged with the alteration of brightness and contrast. This method of image thresholding has been suggested in the Image J user guide (https://imagej.net/docs/guide/). We added this information in the Discussion section (line 805 – line 815).

  1. A second major point concerns the measurement of osteocyte and lacunae size via the etching method as shown in Figure 9. It is hard to believe that the black border around the objects measured in the electron microscope are the lacunae surrounding the cells. It is much more probable that these are “shadowing” effects arising from the elevated shape of etched lacunae. Furthermore, the black area in the middle of what the authors call “osteocyte” is another indication that “black” in these images cannot be attributed to lacunar space. See also Figures 2C in Tiede-Lewis & Dallas, Bone 122, 101 (2019) or Figure 2a, b in Milovanovic et al., ACS Nano 7, 7542 (2013) – these figures also show dark fringes around the osteocytes, but these can hardly be classified as lacunar space.

RE: We thank the reviewer for this valuable comment. There are some different opinions regarding the etching method. For example, in previous publications, researchers used acid-etched resin-embedded samples to obtain images and observed the osteocyte-lacunocanalicular system with these acid-etched SEM images[1, 2].

The measurement of lacunae was performed manually. We only included the black area around the osteocyte as the pericellular area, as seen in Figure 10C and Figure 10F, which showing the location of osteocytes and lacunae for analysis. This method we followed to identify the osteocyte and pericellular area has been used and published previously, as seen in the images presented below[3].

  1. Finally, are these really BSE images as the authors write in the figure caption? It rather looks like secondary electron images.

RE: We thank the reviewer for this question. It is a BSE image. Information could be found in the original data listed below (Det: BSE)

Minor points

1.) Why are areas less than 8 μm2 defined as noise? As 2D sections of lacunae are investigated, there is no lower limit on the possible size of a lacuna cross-section.

Responses: The critical point in establishing the current method is setting a suitable identification range for osteocytes. As we described in our manuscript, to set an identification range, we first tried to calculate the area of all objects in every BSE picture. After the area calculation of all objects, we got the area histogram, including all objects. These objects might include noise objects, osteocytes, and other canals such as blood vessels, Haversian canals, or bone marrow cavities. From observation of the histogram picture, we could find the highest peak (as shown in Figure 3D). From observation of the original images, we also found that osteocytes occupied most of these objects. So, the highest peak should be osteocytes. Therefore, the two side edges of the peak were set as the isolation lines of osteocyte.

2.) What was the criterion to define 280 μm2 as the maximum size of lacunae?

3.) Which sample current was used?

Responses: We thank the reviewer for this valuable comment. The sample sections were coated with gold before observation using BSE. We added this information in the revised manuscript (line 98- line 99, line 252).

4.) Line 91: To which pixel resolution (micrometer/pixel) does a magnification of 200 correspond to?

Responses: The pixel resolution corresponding to the magnification of 200 is 1.35 (micrometer/pixel). We added this to the manuscript revision (line 130- 131).

5.) Line 146: What does “lowest” and “highest level of a circle” mean?

Responses: To make this clear, we have revised the description and used “The Circ. ranges from 0 (infinitely elongated polygon) to 1 (perfect circle)” in the revised manuscript (lines 194- 197).

6.) Why is the “Circ” Parameter used (and how is it defined?) to discriminate lacunae from cracks, when other parameters, like aspect ratio or roundness are used later for the description of morphometric data? Would it not make more sense to use one of these latter parameters to discriminate lacunae from cracks?

Responses: We thank the reviewer for this valuable comment. The Circ. Aspect ratio and Roundness can be used to remove cracks, and we selected Circ. as the filter and tried to calculate Aspect ratio and Roundness as the final results. The definition of Circ. parameters follows the user guide of Image J on the Image J website (https:// imagej.net/docs/guide/).  with a value of 1.0 indicating a perfect circle. As the value approaches 0.0, it indicates an increasingly elongated shape. No evidence supports one parameter is senser to others in this statement.

7.) No technical details on microCT measurements are given: pixel resolution, exposure time, number of rotations, energy, ...

Responses: We thank the reviewer for this valuable comment. We used the micro-CT system (µCT 50; Scanco Medical AG, Bassersdorf, Switzerland). The bones were scanned at an energy of 70 kV and intensity of 114 µA, with 300 ms integration time, resulting in 16-µm isotropic voxel size. And the pixel resolution is 0.016(mm/ pixel).

These details have been added in our revised manuscript revision (line 261-line 265).

8.) Sections 3.1 and 3.2 can be deleted.

Responses: We thank the reviewer for this valuable comment. Sections 3.1 and 3.2 were reserved to show the full results of experiment better.

9.) Line 239: What does “50% and 95% of the area of osteocyte lacunae in the diabetes group decreased compared to the control group” mean? I guess, it refers to the fact that the cumulative frequency distribution of the control is below the diabetes group? But this is valid for the entire curve, not for only two points. Please, rephrase.

Responses: We thank the reviewer for this valuable comment. The sentence is to state that the cumulative frequency distribution of the control group is lower than the diabetes group. This section has been rephrased following your suggestion (line364 –line365, line 369- line 370).

10.) Figure 6: It is stated that panels A and D are box-plots – but no there is no box.

Responses: We thank the reviewer for this valuable comment. This part has been rephrased in our revised manuscript (line360 –line370).

11.) Figure 7: why is the cumulative frequency in this plot (panels C and F) presented differently than panels C and F in figure 6? I suggest to do consistent plots.

Responses: We thank the reviewer for this valuable comment. The data in the two groups showed the only a minor difference. We have tried to use the same line picture as in Figure 6 (panels C and F) to show the cumulative frequency of the two groups, however, the cumulative frequency lines in the two groups almost overlapped with each other and were hard to be distinguished as two lines. So, we presented it in another form, as shown in Figure7 (panels C and F).

12.) Figure 7, Panel B: It is surprising that the distribution of lacunae roundness is so similar for controls and diabetes – even the peak at 0.65 is reproduced. It is hard to believe that this peak is not an artifact. If not, why should the roundness show this behavior?

Responses: We thank the reviewer for this valuable comment. We were actually surprised with these results too, but the roundness still has minor differences in two groups which can be observed in panel B. And this result was based on an automatic calculation by Image J software and included almost 10 thousand data of osteocytes. Our re-analysis (analysis 2) also shows a similar phenomenon (Figure 9 of the revised manuscript). This phenomenon needs further research to be explained, which will be performed in our future study.

13.) Line 298 and following: stay consistent in nomenclature: “F “should be “Figure”.

Responses: We thank the reviewer for this valuable comment. This error has been revised accordingly.

14.) Figure 10: in the plots of this figure 9 data points are given for controls and diabetic group, respectively. According to the section 2.6 these stem from 3 samples per group where again 3 cubes per sample were prepared. So, the figure mixes results from one animal (intra-individual variation) with results from different animals (inter-individual variation). This should not be done. Panel E: Why are the axis interchanged in this panel? In all other panels, diabetes and controls are place on x-axis, the corresponding variable on y-axis.

Responses: We thank the reviewer for this valuable comment. We have adjusted the quantification data analysis by using the nested t test, which considers these biases. And we also changed the figure form to show these data (Figure 11). New results were given in our revised manuscript.

15.) In the reference list important work from other groups dealing with similar approaches is missing, e.g., Milovanovic et al., ACS Nano 7, 7542 (2013), Jandl et al., Bone 135, 115324 (2020), Blouin et al., J. Bone Miner. Res. 32, 1884 (2017) or Mähr et al et al., Int. J. Mol. Sci. 22, 4508 (2022).

Responses: We thank the reviewer for this valuable comment. We have cited these three articles following your suggestion (Reference list, 22,23,35).

References

  1. Feng, J.Q.; Ward, L.M.; Liu, S.; Lu, Y.; Xie, Y.; Yuan, B.; Yu, X.; Rauch, F.; Davis, S.I.; Zhang, S.; Rios, H.; Drezner, M.K.; Quarles, L.D.; Bonewald, L.F.; White, K.E. Loss of DMP1 causes rickets and osteomalacia and identifies a role for osteocytes in mineral metabolism. Nat Genet. 2006, 38, 1310-1315.
  2. Milovanovic, P.; Zimmermann, E.A.; Hahn, M.; Djonic, D.; Puschel, K.; Djuric, M.; Amling, M.; Busse, B. Osteocytic canalicular networks: morphological implications for altered mechanosensitivity. ACS Nano. 2013, 7, 7542-7551.
  3. Lai, X.; Price, C.; Modla, S.; Thompson, W.R.; Caplan, J.; Kirn-Safran, C.B.; Wang, L. The dependences of osteocyte network on bone compartment, age, and disease. Bone Res. 2015, 3,

Reviewer 2 Report

The authors provided a novel and reproducible method for the analytical study of osteocytes/lacunae. 

It is not clear to me the main goal of the paper: to provide a method for the study of osteocytes or to study the effects of diabetes on bone density?

Is this analysis applicable to other diseases affecting the bone? Which kind of benefits willl provide?

It is not clear to me the potential traslational impact of this study. 

Fig. 9 G-H-I: the results reported in the graphs do not seem significative

Author Response

Dear Editors,

Thank you for your letter and for providing us with the reviewers’ comments on our manuscript. Those valuable comments are constructive for revising our paper, providing crucial guiding significance to our research. We have carefully revised our manuscript accordingly and believe that the revised manuscript can meet the criteria for acceptance. The changes are marked in blue and detailed in the point-to-point responses below.

Comments from Reviewer 2:

The authors provided a novel and reproducible method for the analytical study of osteocytes/lacunae. It is not clear to me the main goal of the paper: to provide a method for the study of osteocytes or to study the effects of diabetes on bone density?

Responses to comment 1:

We thank the reviewer for this valuable comment. Our paper includes two goals. The first one is to establish a method for the study of osteocytes. The second one is to use the established method to analyze diabetic bone quality with the measurement of osteocyte/lacunae size in the mandibular cancellous bone of a pig model. We have rephrased the aims in the revised manuscript accordingly. (line 70-76)

Comment 2:

Is this analysis applicable to other diseases affecting the bone? Which kind of benefits will provide? It is not clear to me the potential translational impact of this study. 

Responses to comment 2:

We thank the reviewer for this valuable comment. This analysis is applicable to bone associated physiology and pathology, especially in the evaluation of osteocytes/lacunae state to suggest whether bone quality is affected or not. Further studies will be performed to refine the method as a diagnostic tool in the future.   

Referees’ comment 3

Fig. 9 G-H-I: the results reported in the graphs do not seem significative.

Responses to comment 3:

We thank the reviewer for this valuable comment. We refined the graphs in our revised manuscript to show the results (Figure 10G, 10H, 10I of the revised manuscript) and the methods of analysis are listed below. Analysis results suggest the difference. Figure 10G: The size of osteocytes decreased significantly in the diabetes group (p=0.0003<0.001, Mann-Whitney test); Figure 10H: The lacunae area was considerably reduced in the diabetes group (p=0.0002<0.001, Mann-Whitney test); Figure 10 I: The pericellular area was reduced to a small extent in the diabetes group, this difference also has a statistical significance (p=0.0164<0.005, Mann-Whitney test).  

Round 2

Reviewer 1 Report

My comments can be found in the attached pdf-file.

Author Response

Dear Editors,

Thank you for your letter and for providing us with the reviewers’ comments on our manuscript. We have carefully revised our manuscript accordingly and believe that the revised manuscript and our rebuttals to the reviewer’s queries can be satisfied with the reviewer and meet the criteria for acceptance. The changes are marked in blue in the revision and detailed in the point-to-point responses below.

1.) I am still extremely doubting that using the etching method the authors can reliably measure lacunae area (Figure 10 in the manuscript). In favor of their claim the authors were referencing the paper of Lai et al., Bone Research 2015, but in this paper the analysis was done using transmission electron microscopy (TEM), which is a completely different story than using scanning electron microscopy (SEM) techniques. Furthermore, I repeat my question if the pictures were taken in secondary or backscattered mode? In the other paper that the authors are citing (Milovanovic et al., ACS Nano 2013) a similar method as in the current manuscript is used (albeit in this paper lacunar area is not measured, only number of canaliculi emanating from each osteocyte), but Milovanovic et al. report on images taken in secondary mode (which seems to be the more appropriate setting for such an investigation).

RE: In our previously published papers [11, 17], we demonstrated that backscattered SEM and acid etching methods will enable the analysis of osteocyte morphological changes and lacuna-canalicular network in resin-embedded bone samples. Using the acid etching method, we have also shown the changes in mineral density around the osteocyte body, which have established the foundation for the techniques developed in this paper to measure the lacunae area. The acid will etch the minerals around the canaliculi to enable the view of the canaliculi network. Similarly, the acid will also etch the minerals around the osteocyte body. Due to the progressing decrease of mineral contents close to the osteocyte body, the lacunae area can be pictured as shown in Figure 10 in this study with the appropriate acid concentration and etching time. Compared with the cell body, dendrites, and high-density bone matrix, demineralized areas around the osteocyte body, the lacunae area, will show a dark zoom under the backscattered SEM. 

In this study, we reported, for the first time, measuring the lacunae area using the acid etching method and backscattered SEM. We believe that our study will give some information on how to analyze osteocyte/lacunae morphological changes. Using a diabetes disease model, we validated the measurement on osteocytes and lacunae. We have added in the discussion section to clarify the methods for lacunae area measurement (line 917 – line 932).

[11]   Jaiprakash, A.; Prasadam, I.; Feng, J.Q.; Liu, Y.; Crawford, R.; Xiao, Y. Phenotypic characterization of osteoarthritic osteocytes from the sclerotic zones: a possible pathological role in subchondral bone sclerosis. International journal of biological sciences. 2012, 8, 406-417.

[12]   Du, Z.B.; Ivanovski, S.; Hamlet, S.M.; Feng, J.Q.; Xiao, Y. The Ultrastructural Relationship Between Osteocytes and Dental Implants Following Osseointegration. Clin Implant Dent R. 2016, 18, 270-280.

2.) The authors write that Circ and Roundness are equally suited for discriminating lacunae from cracks. In this case I would advice the authors to skip the Circ parameter and use Roundness. Of course this is mainly due to aesthetic reasons (and for sake of simplicity for the reader, as fewer parameters make the manuscript easier to read) and I do not insist on this point, if entire evaluations would have to be re-done.

RE: We thank the reviewer for this valuable suggestion. However, all parameters selected in this study are to justify our claims as for the first image study to define osteocytes and their changes in bone samples embedded in resin. Combining the parameters of Circ and Roundness will enable us to distinguish the lacunae from small artifacts due to cracks.

3.) In section “2.8 Statistical Analysis” the authors write that normal distributed data are presented as Mean +/- SD, while non-normal distributed data are presented as box-plots median with interquartile range. But in Figure 6, 7 and 9 all median/interquartile range data are not given as boxplots.

RE: We thank the reviewer for pointing this out. Figure 6, 7 and 9 have been revised accordingly.

4.) Figure 7, Panel B: It has to be clarified, where this peculiar behavior in Roundness for values around 0.65 stems from. I would guess that it is some artifact due to binning? Also in the histogram shown in Figure 3 there seem to be some peculiarities due to binning, as for small areas the data between adjacent bins are hopping between values of around 100 and zero.

RE: We thank the reviewer for this valuable comment. We adjusted the binning of Panel B in Figure 7. For Figure 3, we used the original histogram to replace the picture of Panel C in Figure 3. We also gave explanations about the range setting for lacunae/osteocytes based on this original histogram in Materials and Methods sections (line 186 – line 225).

5.) Figure 7, Panel B and E: as aspect ratio is just the inverse of roundness, why are both parameters given? One is sufficient. Coming back to my previous statement in 2, a reduced number of parameter increases the readability of the manuscript a lot.

RE: For the actual image analysis, the definition of these parameters will be more precise to measure the particle changes. The aspect ratio shows that the most osteocyte lacunar are spindle shape. This is to follow the user guideline of Image J program, from which aspect ratio and roundness are calculated based on the following formulas:

Aspect ratio: The aspect ratio of the particle’s fitted ellipse, [Major Axis]/ [Minor Axis]

  or the inverse of Aspect Ratio

Therefore, two parameters were measured and presented in the study.

6.) The two paragraphs in the discussion section starting with “Global thresholding is an image ….” (lines 515 – 538) should go into the Materials and Methods sections as it is not appropriate for the discussion. Concerning the statement “The pixel values are unchanged with the change of brightness and contrast.” I would be careful, as I can easily think of situations where this is not true, e.g., in the case of over-saturation when all pixels are white. The statement of the authors most probably refers to “sensible” settings of brightness and contrast.

RE: We thank the reviewer for this valuable comment. We moved these paragraphs into Materials and Methods following the suggestion (line 148 – line 166).

As we used “intermodes global” thresholding (Figure 2 B) to identify the intensity values of the original BSE image, the histogram of image intensity value was given by image J software. In a sensible setting of brightness and contrast, the pixel greyscale histogram of all included original BSE image appears typical two peaks. In the case of over-saturation when all pixels are white, the greyscale histogram will not appear typical two peaks, and cannot be included in our research. This ensures the original BSE image has sensible range of brightness/contrast quality, and thereby ensures the thresholding. We added this setting in our revised manuscript (line 159 – line 166).

Reviewer 2 Report

The authors answered all my quesyions

Author Response

We thank the reviewer for the commitment of our work.